# Proposal of RBPA: Retrieval-Augmented-Generation based Personal Investment Assistant

**Xue Zeng**[*]
PBC School of Finance
Tsinghua University
Beijing, China
zengx24@mails.tsinghua.edu.cn

**Yinuo Li**[*]
Department of Construction Management
Tsinghua University
Beijing, China
liyinuo24@mails.tsinghua.edu.cn

## 1 Introduction

### 1.1 Background

Large language models (**LLMs**) have made significant strides in recent years, with their capabilities being harnessed across a multitude of industries for various applications. These models, with their vast parameter counts, are designed to handle complex tasks and data, offering enhanced expressiveness and predictive performance.

Although LLMs can grasp basic world knowledge, they cannot be directly applied to financial markets in dynamic games. The influencing factors of the financial market are complex, from macro to micro and involve a wide range of aspects, and the analysis of the financial market needs to establish a professional knowledge base in the financial field, including basic professional knowledge, logical chain knowledge, and related network knowledge.

Retrieval-Augmented Generation (**RAG**) is a groundbreaking approach that marries the capabilities of large language models with the precision of information retrieval from reliable database. By leveraging a vast knowledge base, RAG enables the generation of highly accurate, relevant, and timely responses, making it an ideal technology for a personal investment assistant, under which circumstances lots of professional data and personal data are required.

### 1.2 Related Works

**RAG**   Related works in the field of RAG have focused on enhancing the capabilities of large language models (LLMs) by incorporating external knowledge databases. Efforts like LangChain[1] and OpenAI's text-embedding-3 model[2] laid the foundation for vector-based retrieval systems. Works such as the introduction of Palm by Chowdhery et al.[3] and the Mistral 7B model by Jiang et al.[4] also pushed the boundaries of scaling language models. Recent advancements in RAG systems, like Piperag by Jiang et al.[5], have emphasized the importance of fast retrieval-augmented generation. Meanwhile, studies on efficient vector search[6][7] and cache management strategies[8][9] have aimed to optimize the performance of RAG workflows. Notably, the work of Kwon et al.[10] on PagedAttention has significantly improved memory management for LLM serving which is very important for RAG system for it usually takes very long inputs. Recently, Darren et al.[11] implemented graphRAG, which use LLM for database building and retrieve instead of traditional vector database and vector search.

**AI in investment**   Kim and Nikolaev[12] exploit a artificial neural network combined with a large language model (BERT) to model the interpretation process in a way that allows us to directly capture the value of interactions between multimodal data. Citadel, a multinational hedge fund and financial

---

[*]These authors contributed equally to this work.

services company with USD62 billion in assets under management, aims to include ChatGPT in its operations[13]. LLMs can discern intricate details from earnings reports to macroeconomic studies and process vast amounts of unstructured data, such as news articles or expert opinions, more efficiently than human analysts[14]. However, none of them can give personal investment suggestions for investors based on comprehensive information.

Therefore, to fill the gaps in the investment assistant field, we are going to present RBPA, a new system using RAG to help get professional data and personal data to generate better and more responsible suggestions for a investor.

## 2 Method

In order to help our system to gain the ability to generate responsible suggestions for a certain investor, we mainly need to do two jobs:

- build external database to get advanced knowledge of finance and the real-time information, as well as basic situation of the investor the system is now serving;
- help the LLM to get stronger logical ability on this specific field.

### 2.1 Graph database for RAG

Considering traditional vector database normally just split each document into several chunks, and saving the embedding of each chunk as the key of their corresponding text. This method of building vector database will easily lose many valuable information if they are far from each other. So, in order to keep long distance relationships and try to capture all useful information in each document, we will use graph database which contains different instances and relationships we collected from the documents. This process is designed to be done using LLM, with carefully designed prompt or some finetune if necessary. After the graph is built, we will use certain ways to merge similar information into groups, and use LLM to summarise each group in order to process multi-layer fast retrieve during inference.

### 2.2 Ability to deep-analyse finance market

In order to help the LLM to gain stronger ability to deeply analyse the market, we designed to try two different ways to see which works better. First is a easier way, which use high quality analyses from human in the external database for "logical support", and will be extract from the database during the retrieve as well; the second way is using these data to finetune the LLM if the first way doesn't work well enough.

As for data collection, we planned to collect several types of data from the real world, including price data, news data, laws and regulations data, financial report and analyst report data, etc. If we are doing the finetune, we will use LLM to collect QA pairs from certain data we need.

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
