# OpenReview forum: "【Proposal】RBPA: Retrieval-Augmented-Generation based personal investment assistant"
_tsinghua.edu.cn/THU/2024/Fall/AML — THU 2024 Fall AML Submission_

### Official Review · ~Ethan_Wei_Yuxin1 · 2024-11-08
**The good and the bad of this paper**

**Rating:** 7
**Confidence:** 4

**Review:**

It is great that this paper considers using varied data sources and alternate data sources like price data, news, laws and regulation on top of financial and analyst reports. RAG is a great way to ensure high reliability from a low likelihood of hallucinations of an LLM. However, I feel that the technicalities on how the paper plans to introduce "logical support" could be further expanded upon. If a RAG system is in use, I'm not too sure if fine-tuning should be part of the process as well, since that may defeat the purpose of a RAG system which allows for vector database retrieval without needing to train an LLM on the new data.

---

### Official Review · ~Peidong_Zhang1 · 2024-11-08
**Strengths and limitations of proposal**

**Rating:** 7
**Confidence:** 4

**Review:**

This proposal presents RBPA, a RAG-based system for personalized investment advice, leveraging external financial data and enhancing LLM capabilities through a graph database and fine-tuning. The approach of using a graph database for richer information retrieval is a key strength, offering the potential for more nuanced financial recommendations. However, the proposal lacks details on performance evaluation, data privacy concerns, and the scalability of the system in dynamic real-world scenarios. These gaps need to be addressed to strengthen the feasibility and impact of the proposed system.

---

### Official Review · ~Ziang_Zheng1 · 2024-11-09
**an innovative approach to improving large language model (LLM) performance in financial markets through a Retrieval-Augmented Generation (RAG)-based system**

**Rating:** 7
**Confidence:** 3

**Review:**

The paper presents an innovative approach to improving large language model (LLM) performance in financial markets through a Retrieval-Augmented Generation (RAG)-based system, named RBPA. This system aims to generate personalized and data-informed investment advice by combining graph databases with LLMs. The methodology is well-conceived, and the proposal is timely given the growing intersection of AI and finance. However, there are several areas where the paper could benefit from refinement and additional clarity.

### Strengths

1. **Relevant Problem Statement**: The paper clearly identifies the complexity of financial markets as an application domain for LLMs. The focus on creating a personalized investment assistant via RAG is compelling and addresses a real-world need.

2. **Novel Approach Using Graph Databases**: Introducing a graph database to retain long-distance relationships between data points is a notable innovation. This addresses the limitation of vector databases that often struggle to capture context across widely separated text segments.

3. **Flexible Strategy for Model Training**: The paper’s dual approach to enhancing the model’s analytical abilities—either using external high-quality analyses or fine-tuning the LLM if needed—demonstrates an adaptable methodology that aligns well with model performance requirements.

### Areas for Improvement

1. **Real-Time Data Management**: The financial market is highly dynamic, with constantly changing data. A more detailed explanation of how the graph database will be updated and maintained in real-time would greatly improve the proposal. Readers would benefit from a discussion on the methods and technologies for ensuring timely, accurate data updates.

2. **Data Quality and Bias**: The system’s reliance on varied data sources such as news articles, financial reports, and regulations raises potential concerns about data consistency and quality. Clarifying how the model will handle and mitigate biases in these sources could enhance the paper’s rigor, especially given the impact of biased data on investment recommendations.

3. **Comparative Analysis with Existing Systems**: Although the paper mentions recent advancements in RAG systems and industry applications (e.g., Citadel’s use of ChatGPT), a more thorough comparison between RBPA and similar existing systems would strengthen the contribution. Highlighting specific challenges unique to the financial industry and how RBPA addresses them would provide a clearer value proposition.

4. **Scalability and Performance Evaluation**: Given the complexity of RBPA’s data pipeline, details on the system’s scalability and potential bottlenecks would be valuable. Additionally, the authors could consider including preliminary benchmarks or plans for evaluating the system’s performance, both in terms of retrieval speed and recommendation quality.

5. **Data Privacy and Personalization**: Since the system integrates personal data with professional financial information, a discussion on data privacy considerations is essential. This includes methods for securing personal information, especially if the system is deployed in real-world settings where regulatory compliance may be required.

### Recommendation

The paper has potential as it addresses a significant problem with a novel approach. Addressing the above concerns will make it more robust and applicable to real-world financial applications. I recommend acceptance with major revisions, especially focusing on real-time data management, bias handling, and performance evaluation. Expanding on these areas will help make RBPA a stronger and more competitive solution in the AI-driven finance landscape.

**Suggested Action**: Accept with Major Revisions

---

### Official Review · ~Chan_Thong_Fong1 · 2024-11-10
**Evaluating the Innovation and Empirical Rigor of RBPA’s Graph-Based Financial Advisory System**

**Rating:** 7
**Confidence:** 3

**Review:**

It is very innovative to use a graph database to enhance RBPA’s ability to capture complex relationships within financial data, thus increasing the model’s precision and relevance in investment guidance. However, the paper would benefit from empirical validation of the RBPA system through backtesting in real-world or simulated market conditions. This could involve measuring recommendation accuracy, response latency, and retrieval precision under various market scenarios, thus strengthening the paper’s claims. Additionally, comparing RBPA's recommendations with those of financial experts or existing investment tools would provide further context for evaluating its effectiveness and value. Overall, with robust empirical evidence, this paper could make a significant contribution to AI-driven investment advisory, offering data-rich, actionable financial support tailored to individual investors.

---

### Official Review · ~Tong_Yu9 · 2024-11-10
**Review of "Proposal of RBPA: Retrieval-Augmented-Generation based Personal Investment Assistant"**

**Rating:** 8
**Confidence:** 4

**Review:**

Quality
The quality of the work is commendable, as it combines established techniques in natural language processing with innovative applications in finance. The authors provide a clear framework for the proposed RBPA system, detailing the need for both external databases and enhanced logical capabilities in LLMs.

Clarity
The paper is generally well-structured and easy to follow. The introduction effectively outlines the problem and the significance of the proposed solution. However, some sections could benefit from more detailed explanations, particularly regarding the implementation of the graph database and the methods for enhancing LLMs' analytical capabilities.

Originality
The approach of combining RAG with LLMs for personalized investment advice is original and timely. While there have been previous works on LLMs in finance, the specific integration of RAG to address the complexities of personal investment is a novel contribution to the field.

Significance
This work holds significant potential for advancing the use of AI in financial decision-making. By addressing the limitations of current systems and proposing a method that combines professional and personal data, the RBPA system could enhance the accessibility and effectiveness of investment advice for individual investors.

Pros
Innovative Approach: The integration of RAG with LLMs for investment advice is a fresh perspective in financial technology.
Addressing Real Needs: The system aims to fill a gap in personalized investment recommendations, which is crucial for individual investors.
Strong Theoretical Foundation: The paper builds on existing research in RAG and LLMs, providing a solid theoretical basis for the proposed system.

Cons
Implementation Details: The paper lacks in-depth discussion on the practical implementation of the graph database and the methods for enhancing logical reasoning in LLMs.
Experimental Validation: There is insufficient information on experimental results or case studies that validate the effectiveness of the proposed system.
Data Collection Concerns: The methods for ensuring the accuracy and timeliness of the data collected for the external database are not clearly outlined.

---

### Official Review · ~Guanglei_He1 · 2024-11-11
**Good proposal, but it lacks depth.**

**Rating:** 8
**Confidence:** 3

**Review:**

The proposal introduces an idea but does not provide a clear approach for validating it. I understand that the core issue here lies in establishing a benchmark dataset. Without a way to quantify this problem, it will be challenging to achieve a meaningful solution.

The key concept of how to use RAG is not clearly explained, especially regarding the current limitations and technical challenges of RAG. Do these technical shortcomings make it very difficult to achieve the proposal’s objectives?

Overall, the issue is that the proposal is overly generic and lacks depth. Wouldn’t applying this concept to personal assistants or knowledge-based Q&A yield similar results?

---

### Official Review · ~Wenjing_Wu1 · 2024-11-11

**Rating:** 7
**Confidence:** 3

**Review:**

**Summary**:

The proposal introduces a Retrieval-Augmented Generation (RAG)-based method to support personal investment decision-making. A notable innovation in this approach is the construction of a graph database tailored to the investment domain, enhancing data organization and contextual understanding.

**Strengths**:
- Well-Structured: The proposal is organized clearly which makes it easy to understand.
- Adequate Background Research: It provides sufficient background information, establishing a strong foundation for the proposed method.
- Innovative Idea: The use of a graph database specifically designed for personal investment is a creative and promising concept

**Weaknesses**:
- Lack of Details in Graph Database Construction: The proposal lacks specifics on how the graph database will be constructed
- Unclear Performance Evaluation Metrics: It’s unclear how the performance of the proposed method will be evaluated. Outlining measurable evaluation criteria would strengthen the proposal by providing a way to assess its effectiveness.

---

### Official Review · ~Yang_Ouyang2 · 2024-11-11
**Innovative and Clear but lack conciseness in technical details**

**Rating:** 9
**Confidence:** 4

**Review:**

Strengths
Relevant: The proposal addresses an important gap in the financial sector of providing personalized, data-driven investment insights.
Clear Methodology: The outlined approach is clear.
Innovative Use of RAG: Using RAG and a graph database to preserve complex relationships between data points is a refreshing take

Weaknesses
Vague Details: The proposal lacks implementation details of the graph database and LLM fine-tuning
Lacks Privacy Considerations: Countries have data privacy and compliance laws.
Absence of evaluation plan.

Overall, the proposal presents an innovative approach to enhancing financial market analysis using RAG and LLMs but would benefit from clearer technical details, data privacy considerations, and an evaluation plan.

---

### Official Review · ~Jiaxiang_Liu7 · 2024-11-11
**Promising Proposal with Novel Approach**

**Rating:** 7
**Confidence:** 4

**Review:**

This proposal outlines the development of a personal investment assistant, RBPA, leveraging Retrieval-Augmented Generation (RAG) to combine large language model capabilities with real-time and domain-specific data retrieval. By utilizing a graph database for capturing complex relationships within financial data and enhancing the model’s logical analysis in financial contexts, RBPA aims to provide more accurate and personalized investment insights. The approach is well-motivated and technically sound, though it would benefit from clearer implementation details and metrics for assessing performance. Overall, this proposal is promising and has the potential to bridge important gaps in personalized financial advice systems.

---

### Official Review · ~Liu_Yiyang1 · 2024-11-11
**Innovative idea that could use more details**

**Rating:** 7
**Confidence:** 3

**Review:**

Generally, the problem's background is well established, and the prospect of using RBPA is promising. Using a graph database to capture long-range relationships between financial data points marks a departure from traditional vector-based retrieval, potentially enhancing the depth and relevance of retrieved information. However, when it came to the methodology, a lot of details were vague. For example, how exactly will the graph database capture complex financial relationships, organize data efficiently, and support fast retrieval? What are the "certain ways" mentioned used to classify information in the last sentence of section 2.1? Evaluation metrics might need a little more details as well, as that will be pivotal to discern when to switch methodologies as mentioned in the first paragraph of section 2.2.

---

### Official Review · ~Gangxin_Xu1 · 2024-11-12
**Review of "RBPA: Retrieval-Augmented-Generation-Based Personal Investment Assistant"**

**Rating:** 9
**Confidence:** 5

**Review:**

The RBPA proposal creatively combines a RAG model with a graph database to enhance financial data retrieval, offering potential advancements over traditional systems. The approach is well-motivated, addressing the need for personalized investment support by capturing complex data relationships. However, the proposal would benefit from more technical details and a clearer evaluation plan.

Strengths:
Innovative Database Use: Leveraging a graph database for contextual data retrieval is a strong enhancement over vector-based systems.
Relevant Problem and Adaptable Approach: The proposal clearly identifies the complexity of personalized investment advice and adapts to performance needs through flexible data handling.

---

### Official Review · ~liyingxin1 · 2024-11-12
**Should add more detail about the background to illustrate why this task is suitable.**

**Rating:** 7
**Confidence:** 3

**Review:**

Investment is a very interesting but random field. It is a special task, so maybe it is very good to use this method and maybe not. The reason to choose it should be clarified. Also, it is suggested to elaborate on the sources and methods of data collection, especially in terms of acquiring and processing financial data, to ensure the reliability and timeliness of the data.